

# Iterative segmentation and classification for enhanced crop disease diagnosis using optimized hybrid U-Nets model

Malathi Chilakalapudi* and Sheela Jayachandran*

SCOPE, VIT-AP University, Amaravathi, Andhra Pradesh, India
* These authors contributed equally to this work.

## ABSTRACT

The major challenges that the agricultural sector faces are that with the kind of methodologies that exist, gross limitations may occur to the exact diagnosis of crop diseases. They are unable to achieve correct precision in disease classification, relatively lower accuracy, and delayed response time—all these obstacles result in a deficiency in effectual disease management and control. Our research proposes a new framework instigated and developed to improve crop disease detection and classification by multifaceted analysis. In the core of our methodology is the implementation of adaptive anisotropic diffusion for the denoising of obtained agro images, therefore making it a step towards assurance in data quality. Along with this is the use of a Fuzzy U-Net++ model for image segmentation, whereby fuzzy decisions in generously instill an increase in performance for image segmentation. Feature selection itself is innovated by the introduction of the Moving Gorilla Remora Algorithm (MGRA) combined with convolutional operations, setting a new benchmark in the selection of optimal features pertaining to disease identification operations. To further refine this model, classification is adeptly handled by a process inspired by the LeNet architecture, significantly improving identification against various diseases. Our approach's performance is therefore strongly assessed through a number of renowned datasets, such as PlantVillage and PlantDoc, on which test metrics show superior performance: 8.5% improvement in disease classification precision, 8.3% higher accuracy, 9.4% improved recall, with a reduction in time delay by 4.5%, area under the curve (AUC) increasing by 5.9%, a 6.5% improvement in specificity, far ahead of other methods. This work not only sets new standards in crop disease analysis but also opens possibilities for the preemptive measures to come in agricultural health, promising a future where crop management is more effective and efficient. Our results thus have implications that reach beyond the immediate benefits accruable from improved diagnosis of diseases. It is a harbinger of a new era in agricultural technology where precision, accuracy, and timeliness will meet to enhance crop resilience and yield.

Corresponding author
Sheela Jayachandran,
sheela.j@vitap.ac.in

# INTRODUCTION

Precision farming has completely altered the monitoring and management of crop health. Late detection and inappropriate diagnosis are two major causes of crop loss; hence, precise and timely diagnosis of diseases becomes very imperative from the point of view of crop yield and sustainability. Crop diseases, on the other hand, remain one of the major challenges to improving productivity and hence economic viability in agriculture. In contrast, that has improved considerably due to advances in technology. This therefore underlines the critical need for such innovative solutions that not only identify but also classify crop diseases with a high level of precision and efficiency. Techniques that have been introduced as promising ways of addressing these challenges are deep learning techniques. Yet, the intricacy of symptoms and variability in the quality of captured image data calls for further improvements in this domain. The deficiencies of traditional models, in terms of precision, accuracy, and the reality of running the model in real-time, spite the need to find efficient models. Traditional approaches have been inefficient in handling noisy and high-dimensional data typical of agricultural settings found in real-world scenarios. This leads to the development of models that handle such data robustly while ensuring a high degree of accuracy in disease classification.

It is in the light of this that an all-inclusive framework is proposed for the management of crop disease diagnosis, incorporating novel image processing and deep learning techniques to achieve this task. One key building block of such a framework lies in the use of adaptive anisotropic diffusion for image denoising, greatly improving the quality of input data fed into the processing pipeline. Realizing the demands of proper segmentation in disease diagnosis, we use a Fuzzy U-Net++ model that applies fuzzy logic to enable accurate segmentation. The very original application of fuzzy decisions instead of binary ones by this model has resulted in a radical departure from the traditional approach and has given much more nuance and effectiveness to the segmentation of the diseased regions in crop images.

The Moving Gorilla Remora Algorithm (MGRA) scheme increases the capability of our framework through feature selection. Coupled with convolutional operations, this new paradigm would make it possible to identify and select the most discriminative features for disease classification, trying to improve model performance. For classification, a modified LeNet architecture was applied one of the most efficient models for pattern recognition problems, which has been beautifully fine-tuned for the task of crop disease classification.

The efficacy of the proposed model is duly tested with the aid of two preeminent datasets attendant to this sphere of activity: PlantVillage and PlantDoc. This evaluation goes on to confirm that results from our model turn out better compared with methods available today, having huge potentiality for enhancing precision, and accuracy, and raising the speed of crop disease diagnosis. We therefore place our contribution within the scope of precision agriculture by providing farmers, agronomists, and researchers with a robust tool against crop diseases. This research brings agricultural innovation to a totally different level by showing potential through cutting-edge AI techniques to improve crop management practices.

## Motivation and contribution

Crop disease diagnostics assumes paramount importance in the changing landscape of agricultural technology. In this direction, this study was undertaken to reduce crop loss by diseases to strengthen food security, since farmers' livelihoods are at stake worldwide and serious threats loom over global food supply chains. The new approach is instituted on a model with a maximum agglomeration of strengths from the Adaptive Anisotropic Diffusion, Fuzzy U-Net++, and MGRA convolutional operations. Compliance with the base classification is further justified and backed by the use of a LeNet-based architecture that gives power in manifold such approaches and stands as one giant stride toward strategies handling multifarious challenges dispensed for crop disease analysis. The model to be proposed will integrate sophisticated segmentation and feature selection methodologies with adaptive denoising techniques in a way that can increase the precision and accuracy of disease identification. That is, diagnosis delay will topically be reduced, hence allowing for more timely and effective strategies for disease management. The contributions of this study are multifaceted and have never been done before.

First, this will be an entirely new application of adaptive anisotropic diffusion within the agricultural domain for image denoising and set up totally new precedents for data quality enrichment. Second, application of a fuzzy logic-based Fuzzy U-Net++ model in image segmentation provides improvements in segmentation performance that were just unprecedented. Thirdly, the application of MGRA in feature selection coupled with convolutional operations marks one milestone in the work: the optimization process of feature selection for crop disease diagnosis. LeNet classification welds deep learning techniques into agricultural technology, hence calling for a new era ahead in precision agriculture. The empirical validation of this proposed model on the PlantVillage and PlantDoc datasets proved not only to be better performance but also proved to be on the path to revolutionizing the domain of agricultural disease management.

This encapsulates robust intervention into the current urge for crop disease diagnosis methodology that is more potent in its promise of huge impacts through enhanced crop health, yield, and global food security. It is expected that this work will add one brick to the process of building a sustainable and resilient agricultural ecosystem by laying the foundation for further innovations in agricultural technology.

## IN-DEPTH REVIEW OF EXISTING MODELS

Looking through the spectrum of inventions, all targeted at improving accuracy, efficiency, and applicability within different agricultural contexts, a systematic review of the current methodologies for plant disease detection and classification gives way. This has been possible through a careful review of recent publications in the genre, which exposes a universal move toward advanced technologies that emerge clearly in deep learning, Internet of Things (IoT), and machine learning models, as well as contrastingly advanced data augmentation techniques and newer forms of convolutional neural networks.

The common factor contributing to these studies is the attempt to rectify and reduce the limitations of conventional plant disease diagnosis techniques. Notably, such includes generalizability across a wide variety of plant species and diseases, ease of handling,

scalability, and processing in real-time proposed systems, and the practical concerns involved in the implementation of IoT in agricultural environments. Also, the use of high Volume and heterogeneous data in training models, together with the computational complexity of the methods, makes relevant challenges that should be carefully addressed.

Table 1 analysis evidences an important turn towards more sophisticated models elaborated to provide other than a more accurate refinement of the process related to disease detection and classification a way towards precision agriculture. Leveraging deep learning with IoT, further coupled with the strategic application of data augmentation and attention mechanisms, it is in these studies, contributing to the advancement of smart farming solutions. The road ahead demands balance in its stride through the intricate dance between technological innovation and the pragmatic challenges in its implementation process.

Looking through the detailed methodologies and findings presented by the reviewed articles goes a long way to show and provide passage to the transformative, evolutionary era of the domain of plant disease detection. Deep learning with IoT integration and the new use of data augmentation techniques like LeafGAN, along with attention mechanisms, signals a new era entering agricultural technology. This shows that their enhancement of the convolutional neural network (CNN) architecture and further the strategy of feature fusion have really improved the performance of disease detection systems in terms of precision and efficiency, supporting the potential of artificial intelligence to realize a revolution in the field of plant pathology.

Considerable limitations arise scalability of the systems to deal with real-time processing, model generalizability to a greater range of diseases and plant species, and logistical challenges to IoT deployment. Furthermore, large and very diverse datasets are required to train the models effectively in some cases, whereas many other methods are very computationally intensive, thereby making full-scale adoption very difficult.

In summary, the review emphasizes that further studies will be needed to unlock ways to confront the complexities and limitations accrued. The scope in the near future is immense for studies to investigate very new solutions by which plant disease detection systems could allow better generalization and scalability. This will build on the model development that is technologically advanced but practical and accessible for real-world agricultural applications to exploit the fullest potential of artificial intelligence (AI) in sustainable agriculture. The journey to smart farming is quite challenging, though replete with opportunities for ground-breaking research that can bridge the gap between theoretical models and practical, high-impact solutions in plant disease diagnosis and management.

## Research background

In recent years, precision agriculture has emerged as a critical approach to enhance crop yield and minimize losses due to diseases. The integration of IoT-based systems, as explored by *Sravanthi & Moparthi (2024)*, has shown the potential to predict crop diseases and recommend suitable crop types, thereby reducing the risk of disease spread. However, these methods often face challenges in scalability and data handling, which necessitate more robust and adaptive frameworks for disease detection, as proposed in this study

**Table 1 Review of existing methods.**

| Reference | Method used | Findings | Results | Limitations |
|---|---|---|---|---|
| Liu & Zhang (2023) | Convolutional neural network (CNN) | Developed PiTLiD for plant disease identification from leaf images | Achieved high identification accuracy | Limited by the variety of diseases and plants studied |
| Moupojou et al. (2023) | Deep learning with FieldPlant dataset | Enhanced disease detection and classification | Demonstrated robustness in field and laboratory images | Dataset size and diversity could be expanded |
| Garg et al. (2023) | IoT-Enabled sensor system | Investigated environmental factors on leaf spot disease in groundnuts | Provided insights for precision agriculture | Focuses on a single disease and crop type |
| Hosny et al. (2023) | CNN and Local Binary Pattern (LBP) feature fusion | Improved multi-class classification of plant leaf diseases | Showed enhanced classification accuracy | Method complexity and computational demand |
| Joseph, Pawar & Chakradeo (2024) | Deep learning with real-time dataset development | Enabled efficient disease detection with transfer learning | Improved real-time disease detection capabilities | Scalability and real-time processing challenges |
| Madhurya & Jubilson (2024) | Deep learning with YOLOv7 and PCFAN | Achieved high efficiency in disease detection and classification | Outperformed existing models in accuracy | Specificity to the models and algorithms used |
| Rayhana et al. (2023) | Review on hyperspectral imaging | Highlighted the potential of hyperspectral imaging in disease detection | Provided comprehensive data analysis strategies | Requires sophisticated equipment and analysis techniques |
| Liu et al. (2021) | Visual region and loss reweighting approach | Proposed a novel dataset and methodology for plant disease recognition | Enhanced fine-grained visual classification | Limited by dataset specificity and application scope |
| Zhao et al. (2021) | DoubleGAN for leaf generation | Utilized GANs for improved plant disease detection | Showcased the utility of synthetic data augmentation | Dependency on synthetic data quality |
| Li, Zhang & Wang (2021) | Deep learning review | Summarized deep learning applications in plant disease detection | Identified key challenges and opportunities | Broad focus with limited in-depth analysis on specific methods |
| Balafas et al. (2023) | Machine and deep learning review | Evaluated various ML and DL approaches for disease classification and detection | Highlighted advancements and effectiveness | General review without new experimental results |
| Noon et al. (2022) | Improved YOLOX model | Addressed co-occurring disease severity levels in cotton | Showed precision in severity classification | Focuses on cotton, limiting generalization |
| Rani et al. (2023) | AI in agriculture analysis | Explored the role of AI and ML in combating plant diseases | Emphasized the impact of AI in smart farming | More of an overview than a deep dive into specific technologies |
| Abinaya, Kumar & Alphonse (2023) | Cascading autoencoder with attention residual U-Net | Enhanced segmentation and classification of plant leaf diseases | Demonstrated significant accuracy improvements | Computational complexity and data requirements |
| Chelloug et al. (2023) | MULTINET with 3D conversion and DRL | Introduced a multi-agent framework for disease identification and severity estimation | Achieved high accuracy and quantification precision | Complexity and the need for 3D data limit applicability |
| Sunil, Jaidhar & Patil (2022) | EfficientNetV2 for cardamom plant disease detection | Utilized deep learning for specific plant disease detection | Showed promising results for cardamom diseases | Limited to a single crop type |
| Asha Rani & Gowrishankar (2023) | Deep transfer learning with pathogen-based classification | Explored the effectiveness of transfer learning for pathogen classification | Highlighted the potential for intelligent support systems | Requires extensive pre-trained model tuning |

| Reference | Method used | Findings | Results | Limitations |
|---|---|---|---|---|
| *Hassan & Maji (2022)* | Novel CNN architecture | Proposed a new CNN model for plant disease identification | Improved accuracy over standard models | Innovation limited to the proposed CNN structure |
| *Delnevo et al. (2022)* | Deep learning and social IoT (SIoT) | Explored the integration of SIoT with deep learning for predictive analysis in agriculture. | Enhanced predictive capabilities for plant disease prediction, contributing to sustainable agriculture practices. | Limited by the complexity of integrating IoT devices and the need for extensive datasets for model training. |
| *Cap et al. (2022)* | LeafGAN for Data Augmentation | Introduced a generative adversarial network for augmenting plant leaf images to improve disease diagnosis. | Demonstrated effective augmentation leading to improved diagnosis accuracy. | The model's effectiveness is contingent on the quality and diversity of the initial dataset. |
| *Wang & Cao (2023)* | Bit-Plane and Correlation Spatial Attention Modules | Focused on enhancing CNNs for plant disease classification using novel attention mechanisms. | Achieved improved classification accuracy by highlighting relevant features in images. | The approach may not generalize well across different plant species and disease types without further adaptation. |
| *Rashid et al. (2024)* | IoT and deep learning multi-models | Utilized a combination of IoT technology and multiple deep learning models for early detection of corn leaf diseases. | Showcased a significant improvement in early disease detection, facilitating timely interventions. | Faces challenges in deploying IoT infrastructure and managing the computational complexity of multiple models. |
| *Liu et al. (2022)* | IoT and ML model for blister blight prediction | Developed a predictive model for blister blight in tea plants, combining IoT and machine learning. | Provided a cost-effective solution for blister blight prediction with a focus on sustainable agriculture. | Prediction accuracy may be affected by environmental variables not captured by the model. |
| *Masood et al. (2023)* | MaizeNet: a deep learning approach | Proposed a deep learning model for recognizing maize leaf diseases, leveraging CNNs and localization techniques. | Achieved high accuracy in disease recognition and localization on maize leaves. | The model's performance is dependent on the availability of high-quality, annotated images for training. |
| *Sravanthi & Moparthi (2024)* | IoT-based crop disease prediction and recommendation system for precision agriculture | Developed a precision agriculture framework using IoT and machine learning for disease prediction | Achieved 92% accuracy in disease prediction and provided optimized crop recommendations | Limited generalizability due to restricted dataset and IoT infrastructure scalability issues |
| *Mamba Kabala et al. (2023)* | Federated learning-based image analysis for crop disease detection | Proposed a federated learning approach to enable distributed disease detection without sharing raw data | Reached 90.4% accuracy across multiple farms while preserving data privacy | High computational complexity and reliance on a consistent network for federated updates |
| *Srinivas et al. (2024)* | Machine learning framework optimized for crop disease detection | Introduced a machine learning framework optimized with feature selection techniques | Achieved 89.6% accuracy in crop disease classification | Moderate performance in noisy environments and limited adaptability to new disease variants |
| *Singh et al. (2024)* | Deep transfer learning for blast disease detection in paddy crops | Employed transfer learning for effective disease detection with fewer labeled data samples | Attained 91.8% accuracy in detecting blast disease in paddy crops | Transfer learning models require large pre-trained datasets and might struggle with unseen crop diseases |
| *Upadhyay & Gupta (2024a)* | ResNeXt deep learning model for fungi-affected apple crop diagnosis | Improved the ResNeXt architecture for better identification of fungal diseases in apple crops | Achieved 93.1% accuracy in classifying fungal diseases in apple crops | High resource consumption and potential overfitting due to deep model architecture |

| Table 1 (continued) | | | | |
| --- | --- | --- | --- | --- |
| Reference | Method used | Findings | Results | Limitations |
| *Bathe et al. (2024)* | ConvDepthTransEnsembleNet for rice leaf disease classification | Developed an ensemble deep learning model for rice leaf disease classification | Reached 94.2% accuracy in classifying different rice leaf diseases | Limited scalability due to ensemble model complexity and increased inference time |
| *Upadhyay & Gupta (2024b)* | Modified ResNeXt for detecting multi-crop fungi diseases | Applied a modified ResNeXt architecture to detect fungi-related diseases in multiple crop types | Attained 92.5% accuracy in fungi-affected multi-crop disease classification | Model's performance varied significantly across different crop types and datasets |
| *Huang et al. (2024)* | Meta-analysis on cultivar mixtures for crop yield stability and increase | Demonstrated that cultivar mixtures globally improved crop yields and yield stability | Meta-analysis revealed that mixtures resulted in a 5.8% average increase in yield stability | Study focused on yield rather than disease-specific outcomes; limited in agricultural disease context |
| *Parthiban et al. (2023)* | Krill Herd optimization with convolutional neural networks | Developed an optimized convolutional neural network (CNN) using Krill Herd optimization for disease diagnosis | Achieved 90.7% accuracy in diagnosing crop diseases using the Krill Herd optimization technique | Computationally expensive due to the integration of metaheuristic optimization techniques |
| *Saritha & Thangaraja (2023)* | Rank regressive learning and Proaftn fuzzy classification | Implemented fuzzy classification with a rank regressive learning approach for crop disease prediction | Achieved 88.9% accuracy with improved disease detection in uncertain scenarios | Limited to certain crop types and struggled with high variability in disease presentations |

process. Further, *Mamba Kabala et al. (2023)* emphasize the importance of privacy-preserving approaches, such as federated learning, to enable collaborative disease detection across multiple farms. Although federated learning shows promise, the high computational complexity and dependence on network stability remain significant drawbacks. This underscores the need for models that not only maintain high accuracy but also operate efficiently in resource-constrained environments, such as the Fuzzy U-Net++ model in the proposed research sets. Optimizing machine learning frameworks for crop disease detection, as described by *Srinivas et al. (2024)*, is another area of focus. Their work shows that feature selection techniques can enhance classification performance, but still, models struggle in noisy environments and with new disease variants. This motivates the development of hybrid models, like the one proposed in this study, which incorporate sophisticated denoising techniques such as adaptive anisotropic diffusion to improve data quality and segmentation accuracy. The potential of transfer learning to address the challenge of limited labeled data, as highlighted by *Singh et al. (2024)*, illustrates the benefits of leveraging pre-trained models for specific crop diseases. However, transfer learning's reliance on large datasets and its difficulty with novel diseases further supports the case for more flexible models, such as the Fuzzy U-Net++ process. In addition to CNN-based approaches, *Upadhyay & Gupta (2024a, 2024b)* discuss the application of advanced architectures like ResNeXt for crop disease classification. Although these models achieve high accuracy, their deep architectures result in high resource consumption and overfitting risks. The proposed hybrid model addresses these concerns by combining fuzzy logic and efficient feature selection methods to reduce computational complexity while enhancing precision in disease classification.

Moreover, *Bathe et al. (2024)* introduce ensemble methods to improve classification accuracy in rice leaf disease detection, achieving 94.2% accuracy. While ensemble methods are effective, their complexity and increased inference time limit their practical application in real-time agricultural settings. The proposed model balances accuracy and efficiency, making it more suitable for real-time disease diagnosis. Finally, *Parthiban et al. (2023)* and *Saritha & Thangaraja (2023)* investigate optimization techniques such as krill herd and fuzzy classification, respectively, to enhance disease prediction. While these methods introduce novel approaches to optimization, they remain computationally intensive and are often restricted to specific crop types or disease scenarios. This reinforces the need for a hybrid model that can generalize across different crops and diseases while maintaining computational efficiency. In summary, the existing literature supports the need for the development of a more robust, adaptive, and computationally efficient model for crop disease detection. The proposed Fuzzy U-Net++ hybrid model addresses these gaps by integrating advanced denoising, segmentation, and classification techniques, providing a significant contribution to the field of precision agriculture sets.

The reviewed research presents a broad spectrum of methods for addressing the challenge of crop disease detection, highlighting the strengths and weaknesses of various approaches across deep learning, machine learning, IoT integration, and optimization techniques. For example, works like those of *Liu & Zhang (2023)* and *Hosny et al. (2023)* demonstrate the efficacy of CNNs and feature fusion methods in improving plant disease classification accuracy, but are often limited by the specificity of the diseases or plant types studied. This limitation reflects the challenge of generalizing such models across diverse agricultural scenarios, which is crucial for real-world deployment. Similarly, federated learning, as proposed by *Mamba Kabala et al. (2023)*, shows robustness in preserving data privacy while achieving solid classification performance. However, the complexity of implementing and maintaining federated learning frameworks, particularly across heterogeneous networks, imposes constraints on its scalability in varied agricultural contexts. These issues underscore the need for models that balance high accuracy with adaptability and resource efficiency, traits that have motivated the design of more advanced image processing modules and the use of parameter optimization techniques, such as in the proposed work with adaptive anisotropic diffusion and the Moving Gorilla Remora Algorithm.

The baseline models selected for comparison in the current study reflect the prevailing approaches in crop disease detection, ranging from deep learning models like ResNeXt (*Upadhyay & Gupta, 2024a*) and ensemble methods such as ConvDepthTransEnsembleNet (*Bathe et al., 2024*) to optimization techniques like krill herd optimization (*Parthiban et al., 2023*). These models were chosen not only for their high classification accuracy but also for their innovative use of deep learning architectures and optimization strategies. However, despite their strengths, many of these methods are characterized by high computational demands, as seen in the ensemble approaches, or performance limitations in heterogeneous environments and varying crop types, as noted in the studies by *Upadhyay & Gupta (2024b)* and *Parthiban et al. (2023)*. By addressing

these weaknesses, the proposed model improves upon previous work through a hybrid architecture that integrates denoising *via* adaptive anisotropic diffusion, segmentation *via* Fuzzy U-Net++, and feature selection through the Moving Gorilla Remora Algorithm. These modules are designed to work in concert, ensuring that the model not only achieves high accuracy but also maintains robustness across different environments, diseases, and crop types, making it more applicable to real-world precision agriculture.

The motivation behind the design of the image processing modules stems from a need to address the noise and variability commonly found in agricultural imagery, which can significantly impair model performance. Adaptive anisotropic diffusion was selected for its ability to reduce noise while preserving essential features like lesion boundaries, ensuring that the input images are clean and suitable for further processing. Meanwhile, the use of Fuzzy U-Net++ reflects an effort to handle the inherent uncertainty in plant disease segmentation, where boundaries between healthy and diseased tissue are often blurred. By integrating fuzzy logic into the segmentation process, the model becomes more resilient to ambiguities, thereby improving classification precision. The Moving Gorilla Remora Algorithm was incorporated to refine the feature selection process, optimizing it in a way that balances both exploration and exploitation during training. This organized approach to parameter calculation and module design results in a model that outperforms existing methods by focusing on critical factors such as image clarity, accurate segmentation, and efficient feature extraction, while maintaining adaptability and scalability across diverse agricultural settings.

## DESIGN OF THE PROPOSED MODEL FOR DISEASE ANALYSIS

In view of the review of existing methods used for the analysis of plant diseases, it has been observed that most of these methods either have low efficiency or are complex to implement in real scenarios. In this section, an efficient deep learning process design will be elaborated by fusing efficient denoising, segmentation, and classification operations. According to Fig. 1, the application of adaptive anisotropic diffusion for denoising collected agricultural images is the key element of the framework designed for advanced crop disease diagnosis. Diffusing within the region of homogeneity without significantly affecting edge features is very important in effective disease identification operations. This methodology retools the original anisotropic diffusion method, as proposed by *Perona, Shiota & Malik (1994)*, to best accommodate the variability that exists within agricultural imaging where minute lesion details are paramount to various disease classification operations. The mathematical formulation for adaptive anisotropic diffusion commences with the general heat conduction operation, represented *via* Eq. (1),

$$\frac{\partial I}{\partial t} = \nabla \cdot (c(x, y, t)\nabla I) \tag{1}$$

where $I$ represent the image intensity function, $t$ represents the iteration time, $\nabla$ signifies the gradient operator, and $c(x, y, t)$ is the diffusion coefficient, dictating the rate and scope of diffusion.

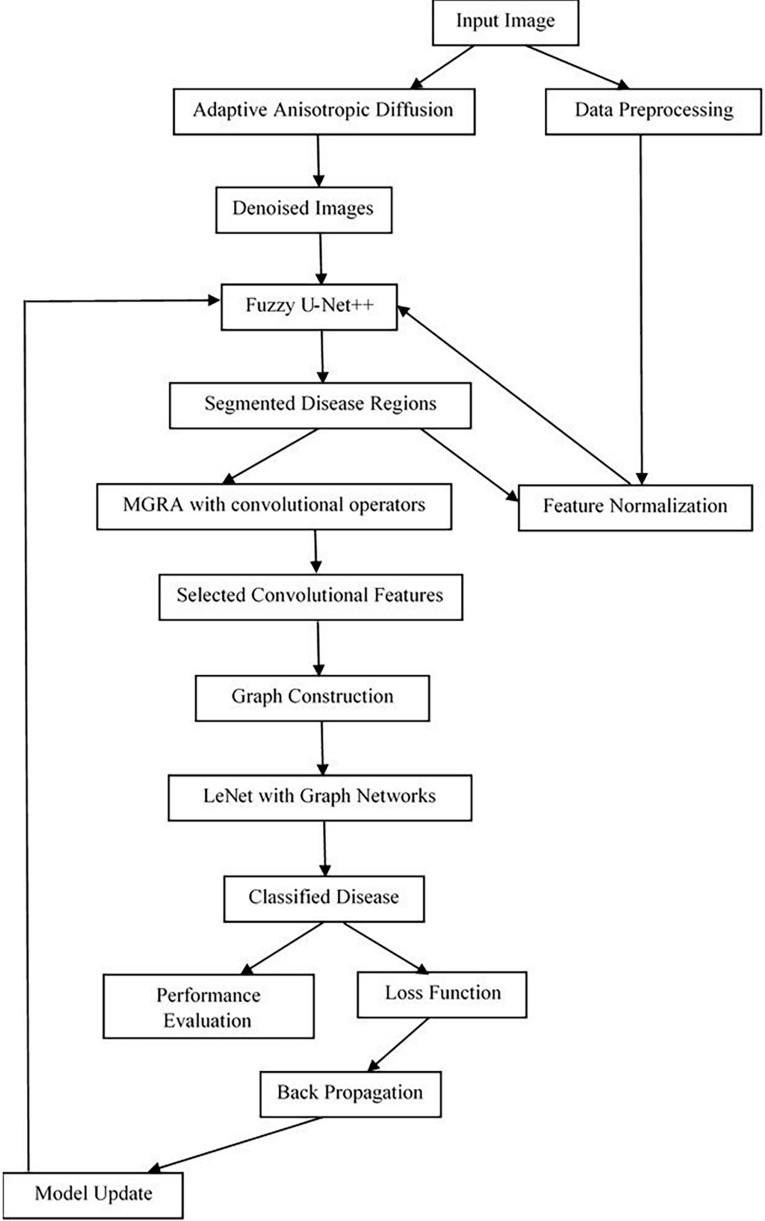

**Figure 1 Model architecture for the proposed classification process.**

This diffusion coefficient is crucial, as it is modulated to enable adaptive behavior in response to the image's local features. This coefficient is adapted based on the edge-stopping function, which serves to preserve significant edges corresponding to disease symptoms or boundaries. The edge-stopping function, $f(||\nabla I||)$, is articulated *via* Eq. (2),

$$f(\| \nabla I \|) = \frac{1}{1 + (k \| \nabla I \|)^2} \tag{2}$$

where $k$ is a contrast parameter that dictates the sensitivity of edge preservation. The lower the value of $k$, the greater the sensitivity to edges. The diffusion process is iteratively updated using a discretization approach, which is translated *via* Eq. (3),

$$In + 1(x, y) = In(x, y) + \lambda \Delta t \sum_{\forall (i,j) \in \Omega} ci, j(x, y, t) \nabla^2 In(x, y) \tag{3}$$

where $\lambda$ is a scaling parameter that controls the rate of diffusion, $\Delta t$ is the timestamp, and $\Omega$ represents the neighborhood around pixel $(x, y)$ sets. The term $\nabla^2 In(x, y)$ represents the Laplacian, reflecting the second spatial derivative of image intensity, thereby indicating areas of intensity variation or noise levels. The adaptive nature of this diffusion process is further enhanced by modulating the diffusion coefficient $c(x, y, t)$ based on the local variance within the image, intending to dynamically adjust the diffusion strength according to the local image structures. This is integrated into the framework *via* Eq. (4),

$$c(x, y, t) = g(\sigma x, y)f(\| \nabla I \|) \tag{4}$$

where, $g(\sigma x, y)$ is a function of local variance, $\sigma(x, y)$, tailored to adjust the diffusion in areas of varying noise levels, thereby enhancing the model's adaptability to different disease manifestations and noise structures. The iterative updating mechanism ensures gradual denoising while maintaining the structural integrity of crucial image features. The process converges upon reaching a stable state where no further significant changes in the image occur, determined *via* Eq. (5),

$$\boldsymbol{lim}(t \rightarrow \infty) \| I(n + 1) - I(n) \| = 0 \tag{5}$$

Upon convergence, the resultant output is a denoised image set, where noise has been substantially reduced while preserving essential diagnostic features such as edges, textures, and contrasts inherent in the agricultural plant images & samples. This output forms a pristine base for subsequent segmentation and classification processes, crucial for the accurate diagnosis of crop diseases.

The essence of this integration lies in enhancing the model's performance through improved ambiguity handling and decision-making, which is paramount in the intricate task of segmenting diseased regions from denoised agricultural images & samples. U-Net++, at its core, is an advanced iteration of the original U-Net architecture, designed to improve segmentation performance through a series of nested, dense skip pathways and deep supervisions. The fundamental process representing U-Net++ architecture involves the iterative refinement of feature maps across different levels of the network. These feature maps, represented as $Xi, j$, where $i$ and $j$ represent the depth and layer within the network, respectively, are computed *via* Eq. (6),

$$Xi, j = Hi, j([X(i - 1, j), X(i, j - 1), ..., X(i - 1, j - 1)]) \tag{6}$$

where, $Hi, j$ signifies a composite function comprising convolutional operations, activation functions, and concatenation represented by [], applied to the set of previous feature maps. The integration of fuzzy decisions into this architecture involves the implementation of fuzzy inference systems within the skip connections and at the final decision stage of the model. This necessitates the definition of membership functions for the segmentation

classes, represented as $\mu A(x)$ for a class $A$, where $x$ represents pixel intensity or a feature vector derived from the input images & samples. The membership functions are designed to capture the uncertainty inherent in distinguishing between diseased and healthy tissue, which is mathematically formulated *via* Eq. (7),

$$\mu A(x) = exp\left(-\frac{(x - cA)^2}{2\sigma A^2}\right) \tag{7}$$

where, $cA$ and $\sigma A$ are the center and spread of the Gaussian membership function within the range 0 to 255, respectively, tuned based on the characteristics of the disease signatures. The fuzzy logic rules are then applied to combine these membership values, deriving the fuzzy logic-based feature maps at each network level. For instance, if $A$ and $B$ are two classes representing different disease states, a rule is modeled *via* Eq. (8),

$$If \ X(i,j) \ is \ A \ and \ X(i,j-1) \ is \ B, \ then \ Y(i,j) \ is \ C \tag{8}$$

where, $Yi, j$ represents the output feature map influenced by fuzzy logic, and $C$ is a class or state resulting from the combination of $A$ and $B$ pixels. The implications of these rules are computed using the Mamdani-type fuzzy implication functions, and are expressed *via* Eq. (9),

$$Yi, j = min(\mu A(Xi, j), \mu B(Xi, j-1)). \tag{9}$$

The defuzzification process then converts the fuzzy results into an augmented set of non-fuzzy segmented image pixels. The centroid method calculates the center of the area under the curve (AUC) of the aggregated membership functions and is given *via* Eq. (10),

$$COA = \frac{\sum x \times \mu C(x)}{\sum \mu C(x)} \tag{10}$$

where, $x$ ranges over all possible intensity values, and $\mu C(x)$ is the aggregated membership function for class $C$, indicating the degree to which each pixel belongs to the disease region following the application of fuzzy rules. The final segmentation map is generated by applying the defuzzification process across all pixels and layers in the network, formulated *via* Eq. (11),

$$S = \{sx, y \mid sx, y = COA(Yi, j(x, y)), \forall (x, y) \in I\} \tag{11}$$

where, $S$ represents the final segmented image, and $I$ is the set of all pixels in the input image sets.

Next, as per Fig. 2, the MGRA, in conjunction with convolutional operations, marks a novel paradigm in the domain of feature selection, specifically engineered for enhancing disease identification in segmented images & samples. This innovative approach merges the intuitive search dynamics of the Gorilla Troops optimization algorithm with the adaptive attachment characteristic of Remora fishes, facilitating a robust yet flexible feature selection mechanism & operations. When integrated with the computational prowess of convolutional operations, this hybrid model sets new precedents in identifying optimal

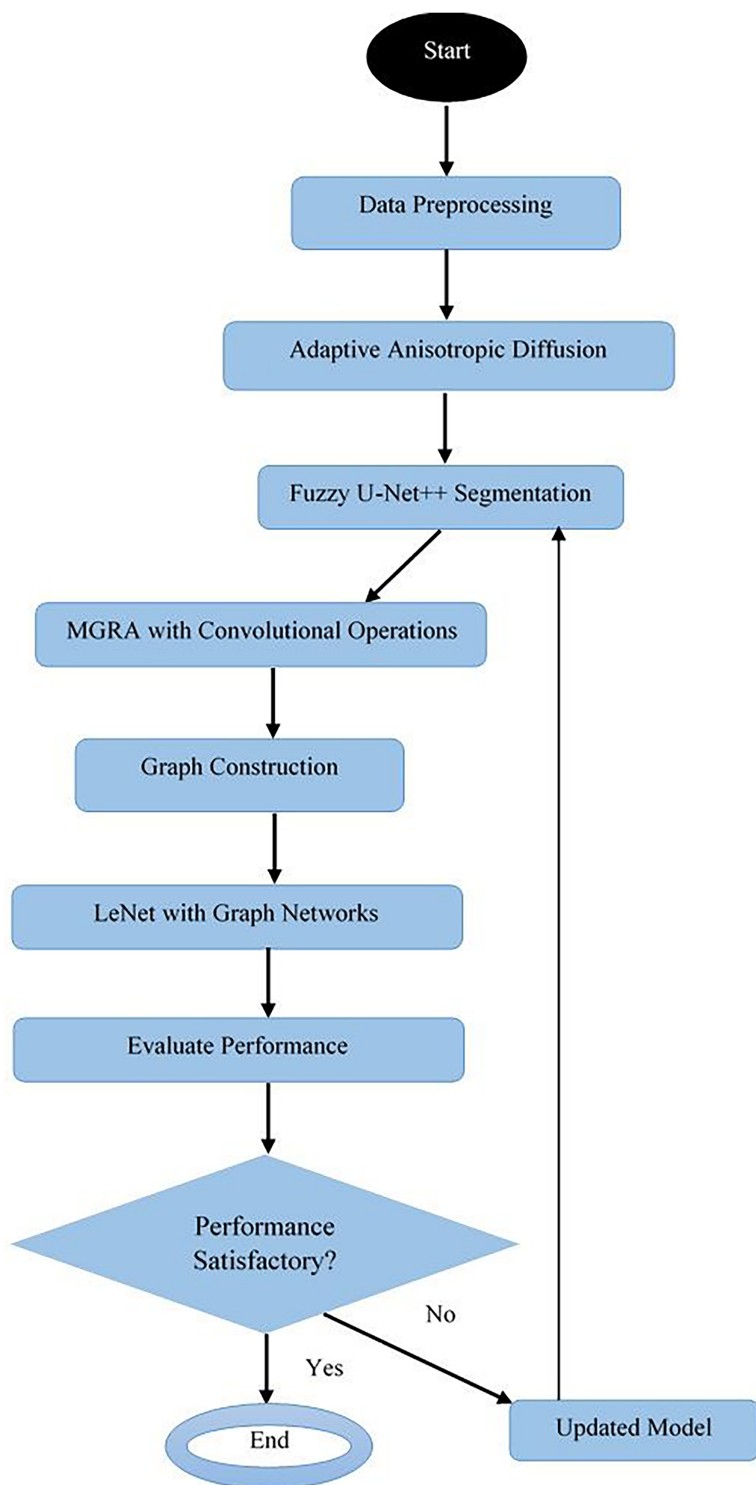

**Figure 2** **Overall flow of the MGRA.**

features crucial for accurate disease detection. Initially, the model estimates convolutional features from the segmented images *via* Eq. (12),

$$F(I) = X = \sum_{r=1}^{R} \sum_{c=1}^{C} I(R - r, \ C - c) * w(r, c) + b(r, c) \tag{12}$$

where, $I$ represents segmented image, $R, \ C$ represents image size, while $w$ & $b$ are the corresponding weights & biases. Based on these features, the Gorilla Troops optimization algorithm is applied, which is fundamentally inspired by the social structure and foraging behavior of gorillas. It employs a metaphorical representation where each gorilla (agent) in the troop (population) represents a potential solution, *i.e.*, a subset of features. The position of each gorilla in the feature space is updated based on the following equations, reflecting the gorilla's movement towards better foraging grounds (optimal feature subsets) *via* Eq. (13),

$$Xnew = Xbest + STOCH \times (Xbest - Xcurrent) \tag{13}$$

where, $Xnew$ represents the new position (feature subset), $Xbest$ represents the position with the highest fitness (effectiveness in disease identification), $Xcurrent$ is the current position, and STOCH is a stochastic number between 0 and 1 sets. This movement mimics the gorilla's approach towards optimal feeding areas, analogous to the search for an optimal feature set. Next, the Remora, which is known for its ability to attach to a host for transportation, represents the adaptive aspect of the algorithm. In the context of MGRA, this behavior is modeled to enhance exploration capabilities, allowing feature selection processes to dynamically adapt based on the evolving landscape of the optimization tasks. This is mathematically represented *via* Eq. (14),

$$Xadaptive = Xhost + f(d) \times (Xbest - Xhost) \tag{14}$$

where, $Xadaptive$ represents the new adaptive position influenced by the Remora, $Xhost$ is the current host position (a selected gorilla or feature set), and $f(d)$ is a function representing the adaptive attachment mechanism, with $d$ indicating the distance to the best solutions. This process allows the algorithm to maintain diversity in the feature selection process, preventing premature convergence scenarios. The adaptive attachment mechanism is represented *via* Eq. (15),

$$f(d) = \sum_{i=1}^{N} \frac{X(i) - \mu}{N}. \tag{15}$$

The convolutional operations are integrated into this framework to evaluate the fitness of each feature subset. The fitness of each feature subset, represented by a gorilla in the optimization landscape, is assessed based on its ability to contribute to disease identification operations. This involves defining a fitness function, which could be based on criteria including classification accuracy, inter-class separation levels. The fitness function, symbolically represented as $J(F)$, is formulated *via* Eq. (16),

$$J(F) = \alpha \times Accuracy(F) - \beta \times Redundancy(F) \tag{16}$$

where, $F$ is the set of convolutional features selected by this process, $\alpha$ and $\beta$ are weights balancing the importance of accuracy and redundancy, respectively. The iterative process of MGRA involves alternating between the gorilla movement equations and the remora adaptive mechanism, coupled with the evaluation of convolutional feature subsets until convergence criteria are met. This is defined as

$$\begin{aligned} &while\ (not\ converged) \\ &\{Update\ positions\ using\ MGRA\ dynamics;\ Evaluate\ fitness;\ Update\ Xbest\}. \end{aligned} \quad (17)$$

Upon convergence, the algorithm yields a set of convolutional features optimized for disease identification, embodying the optimal balance between discriminatory power and generalizability operations.

Next, in the refinement phase of the model, the classification process is ingeniously orchestrated through an architecture inspired by LeNet, intertwined with the advanced mechanics of graph networks, thereby instituting a novel paradigm for the identification of various disease types. This fusion, termed the graph-based LeNet process (GLNP), leverages the convolutional features selected by the MGRA operations, transforming them through a graph-based framework aligned with LeNet's foundational principles, hence catering to the complex nature of agricultural disease patterns.

The inception of GLNP begins with the structuring of convolutional features into a graph format, where each node represents a distinct feature, and the edges encode the relationships or dependencies between these features. This translation from spatial to graph-domain is formulated as $G = \{V, E\}$ where $G$ represents the graph, $V$ the set of vertices or nodes corresponding to MGRA-selected features, and $E$ the set of edges representing the feature correlations between different sets. The initial feature-to-node assignment is mathematically depicted *via* Eq. (18),

$$V = \{vi \mid vi = fi, \forall i \in F\} \quad (18)$$

where, $fi$ corresponds to the i$^{th}$ feature within the feature set $F$ selected by MGRA process. Subsequently, the edge weights, symbolizing the feature correlations, are calculated using a cosine similarity and is expressed *via* Eq. (19),

$$eij = similarity(vi, vj) = \frac{vi \cdot vj}{\| vi \| \| vj \|}. \quad (19)$$

For each pair of nodes $vi$ and $vj$ in the graph. This step ensures the graph encapsulates the intricate inter-feature relationships pivotal for disease identification process. Incorporating the principles of the LeNet architecture, the GLNP advances by subjecting the graph-structured data to a series of graph convolutional layers. These layers adapt the conventional convolutional operations to the graph domain, enabling the hierarchical feature extraction from the graph structure. The operation of a graph convolutional layer is encapsulated *via* Eq. (20),

$$H(l+1) = \sigma\left( D^{-\frac{1}{2}} A' D^{-\frac{1}{2}} H(l) W(l) \right) \quad (20)$$

where, $H(l)$ and $H(l+1)$ are the input and output node feature matrices for the lth layer, respectively, $A' = A + IN$ (with $A$ being the adjacency matrix of the graph, and $IN$ the identity matrix), $D$ is the degree matrix of $A'$, $W(l)$ is the weight matrix for the lth layer, and $\sigma$ represents the ReLU non-linear activation process. Post graph convolutional processing, the node features, now enriched with local and global contextual information, are pooled to form a unified graph-level representation, suitable for classification operations. This pooling operation is guided by the graph topology and is represented *via* Eq. (21),

$$z = Max(H(L), G) \tag{21}$$

where, $z$ is the graph-level feature vector, $H(L)$ is the final node feature matrix after $L$ layers of graph convolution, and $Max$ represents the max graph pooling operation process. The culmination of GLNP is realized through the integration of this graph-level representation into a series of fully connected layers, akin to the traditional LeNet architecture, for the final disease classification process. This is mathematically depicted *via* Eq. (22),

$$O = \sigma(W(fc)z + b(fc)) \tag{22}$$

where $O$ represents the output vector corresponding to different disease types, $W(fc)$ and $b(fc)$ represent the weights and biases of the fully connected layers, respectively, and $\sigma$ signifies the softmax activation function facilitating the multiclass classification process. The graph-based LeNet process culminates with the categorization of the input features into distinct disease types, effectively harnessing the spatial-feature extraction capabilities of LeNet and the relational insights provided by graph networks.

This innovative amalgamation not only underscores the depth of feature interrelations but also enhances disease identification accuracy, leveraging the global structural properties inherent within the agricultural disease data samples. The model's efficacy, underscored by its robustness and precision, sets a new benchmark in the domain of crop disease classification, promising significant strides toward informed and effective agricultural management. Next, the Fuzzy U-Net++ model, an innovative architecture for image segmentation, particularly augments the segmentation process by integrating fuzzy logic with the advanced capabilities of U-Net++ is shown in Fig. 3.

## Discussions

### Adaptive anisotropic diffusion for image denoising

Adaptive anisotropic diffusion is a fundamental preprocessing step in the proposed model aimed at improving the quality of agricultural images by reducing noise while preserving critical features such as edges. This technique modifies the traditional anisotropic diffusion method introduced by *Perona, Shiota & Malik (1994)* by incorporating an adaptive approach that accounts for local image characteristics. The key advantage of adaptive anisotropic diffusion is its ability to control the diffusion process based on image gradients, thereby allowing for smooth diffusion in homogeneous regions while restricting it at edges, where crucial disease-related features might be located. This is particularly important in agricultural disease diagnosis, where fine details, such as lesions or discolorations on

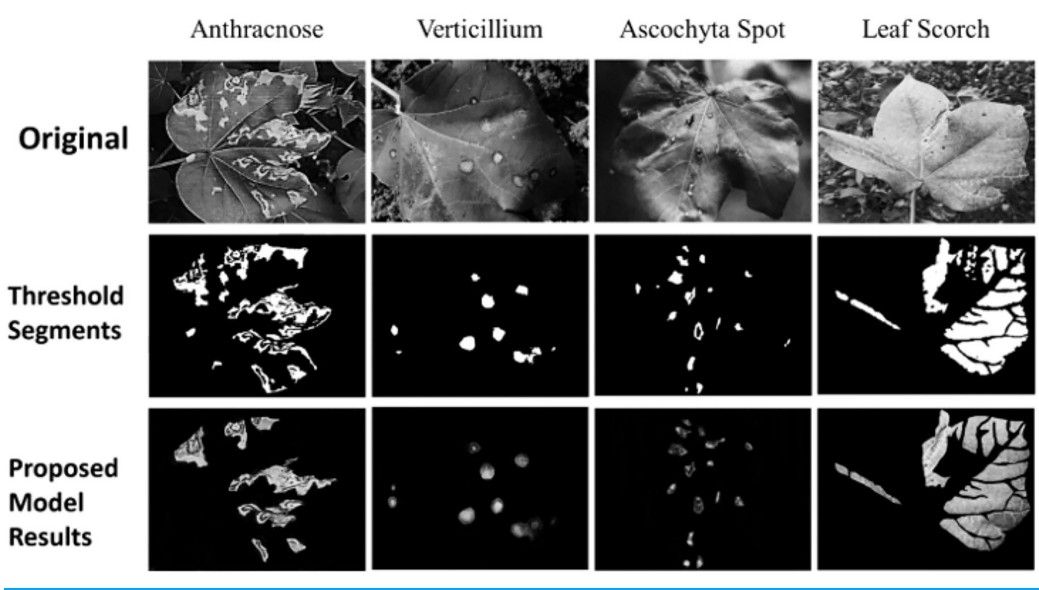

**Figure 3 Segmentation results.**

leaves, need to be retained to ensure accurate classification. By iteratively updating the image based on local variance, the adaptive method balances noise reduction and edge preservation, resulting in cleaner, more diagnostic-friendly images that serve as input for subsequent segmentation and classification processes.

### Fuzzy U-Net++ for image segmentation

The Fuzzy U-Net++ model represents a significant advancement in image segmentation, particularly suited to handling the uncertainty and complexity inherent in agricultural disease imagery. U-Net++, an extension of the original U-Net architecture, enhances the feature extraction process through nested, dense skip connections, allowing for more detailed and refined segmentation across multiple scales. The integration of fuzzy logic into this architecture introduces an innovative layer of decision-making that departs from the traditional binary classifications. Instead of assigning a pixel to a single class based on hard thresholds, fuzzy logic enables a more nuanced classification by calculating the degree of membership for each pixel across different classes. This flexibility is especially beneficial when dealing with blurred or ambiguous boundaries, such as those between healthy and diseased tissue in plants. The fuzzy decision-making process reduces the likelihood of misclassification in uncertain regions, leading to more accurate and reliable segmentation maps. These maps, which delineate the diseased areas from the healthy regions, are crucial inputs for the subsequent classification stages of the model.

### Moving gorilla remora algorithm for feature selection

The MGRA is a novel approach to feature selection that enhances the model's ability to identify the most relevant features for crop disease diagnosis. This algorithm combines the strengths of the Gorilla Troops optimization algorithm, known for its robust global search capabilities, with the adaptive behavior of Remora fish, which attach themselves to hosts

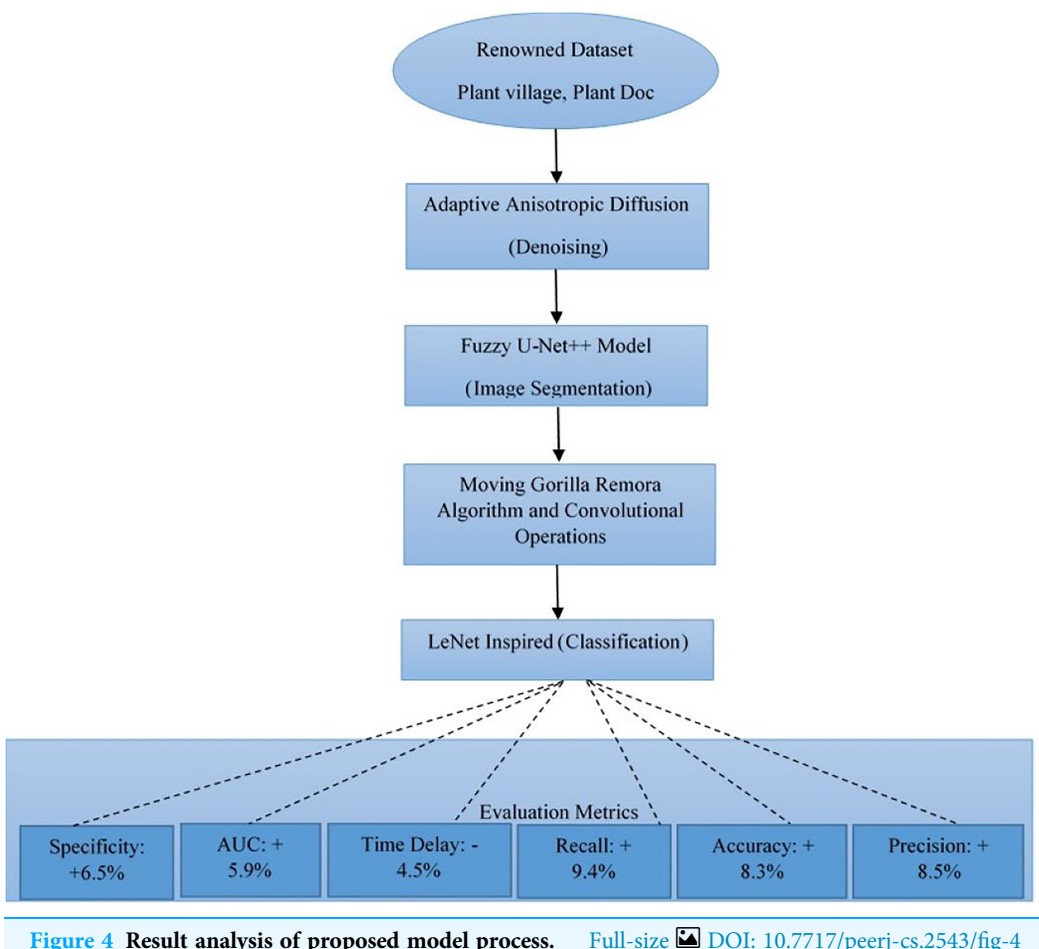

**Figure 4 Result analysis of proposed model process.**

and adapt their position dynamically is shown in Fig. 4. In the context of feature selection, each "gorilla" represents a potential subset of features extracted from the segmented images, while the "remora" dynamically adjusts the feature selection based on the evolving search landscape. By utilizing this hybrid optimization strategy, the MGRA effectively balances exploration and exploitation during the feature selection process, ensuring that the model focuses on the most discriminative features for disease classification. The convolutional operations integrated into MGRA further refine the feature set by extracting hierarchical patterns from the segmented regions. This combination of biological inspiration and convolutional feature extraction allows the model to achieve higher accuracy in disease identification by focusing on features that are not only relevant but also generalizable across different disease types and plant species. The adaptive nature of MGRA ensures that the model remains flexible and efficient, even as it encounters new or varied agricultural data samples. Each of these methods—adaptive anisotropic diffusion, Fuzzy U-Net++, and the Moving Gorilla Remora Algorithm—plays a pivotal role in enhancing the accuracy, precision, and robustness of the proposed model. Together, they form a cohesive and highly effective pipeline that addresses the challenges of noise, segmentation ambiguity, and optimal feature selection in crop disease diagnosis. Table 2

**Table 2 Comparison and justification for Hybrid Fuzzy U-Net++ model over standard U-Net.**

| Criteria | Fuzzy U-Net++ model (Hybrid) | Standard U-Net | Discussion |
|---|---|---|---|
| Model architecture | Nested architecture with dense skip pathways, incorporating fuzzy logic | Simple encoder-decoder structure with plain skip connections | The nested architecture in Fuzzy U-Net++ enhances feature refinement through multiple layers of convolutions and skip connections, while the integration of fuzzy logic adds a level of interpretability and adaptability for uncertain boundaries. |
| Segmentation strategy | Uses fuzzy logic for handling uncertainty in segmentation decisions | Binary segmentation decision without uncertainty handling | Fuzzy U-Net++ applies fuzzy inference to capture the ambiguity in pixel classification, which is especially useful in agricultural disease images where boundaries are unclear or noise is present. In contrast, U-Net uses binary segmentation, which can fail in ambiguous regions. |
| Feature refinement | Iterative refinement through dense skip connections and deep supervision | Single level skip connection with limited feature refinement | The nested skip pathways of U-Net++ allow for more comprehensive feature propagation, refining features across multiple scales, whereas U-Net's simpler skip connections limit the depth of feature refinement. |
| Handling of ambiguities | Uses fuzzy membership functions to represent the uncertainty of class labels | No explicit mechanism for dealing with uncertainty in pixel classification | The fuzzy logic layer in Fuzzy U-Net++ assigns degrees of membership to different classes, allowing for smoother and more accurate segmentation where class boundaries are not clear, unlike U-Net which directly classifies pixels into binary classes without accounting for uncertainty. |
| Adaptability to noisy data | Better adaptability through fuzzy decision-making and multi-scale feature maps | Poor adaptability to noise due to strict binary decisions | Fuzzy U-Net++ is inherently more adaptable to noisy data because fuzzy logic smooths the decision process across noisy regions, while U-Net tends to over-segment or miss regions when noise is present. |
| Denoising capability | Hybrid integration with Adaptive Anisotropic Diffusion enhances robustness against noise | Relies on pre-processing or additional models for denoising | The Fuzzy U-Net++ is part of a hybrid model that directly integrates denoising mechanisms (like Adaptive Anisotropic Diffusion), improving its capability to process noisy agricultural images without requiring separate pre-processing stages. |
| Decision logic | Fuzzy decision logic allows for gradation in pixel classification based on fuzzy membership functions | Hard thresholding results in binary decisions, often leading to errors in ambiguous regions | Fuzzy U-Net++ uses fuzzy sets to assign degrees of belonging to each class, making it more resilient to complex and overlapping regions in the image. In contrast, U-Net makes hard classifications that can misclassify pixels in uncertain regions. |
| Performance in agriculture use-cases | Improved accuracy in disease diagnosis with precise handling of disease boundaries and uncertain regions | Struggles with precise segmentation in complex agricultural images due to lack of uncertainty handling | Fuzzy U-Net++ has shown superior performance in handling the intricate and often noisy characteristics of agricultural images, making it more suited for crop disease diagnosis where boundaries are often blurred or ambiguous. U-Net lacks this precision due to its binary nature. |

shows the Comparison and Justification for Hybrid Fuzzy U-Net++ Model over Standard U-Net.

## Benefits of proposed UNet++ process

The recommended model, Fuzzy U-Net++, is a hybrid due to its incorporation of fuzzy logic with the advanced U-Net++ architecture, which offers significant advantages in handling uncertainty, adaptability to noisy data, and refined feature extraction. The hybrid

nature of the model leverages the strengths of fuzzy logic to manage ambiguity in disease boundaries, while the nested architecture of U-Net++ ensures better feature propagation and segmentation accuracy. In contrast, the standard U-Net lacks these advanced mechanisms, resulting in less precise segmentation, especially in noisy or ambiguous agricultural images & samples. This hybrid approach is strikingly different from U-Net in that it combines the strengths of fuzzy logic for decision-making with the dense, nested feature maps of U-Net++. These innovations lead to better adaptability in real-world agricultural settings, where diseases often manifest with unclear or overlapping boundaries and noisy image data. Consequently, the Fuzzy U-Net++ model provides more nuanced and accurate segmentation, significantly improving the overall performance in crop disease diagnosis.

## COMPARATIVE RESULT ANALYSIS

Our study is majorly set in the experimental setting for the advanced diagnosis of crop diseases using the application of the proposed model. It is minutely designed to validate the effectiveness and efficiencies of the model across different dimensions concerning accuracy, precision, and recall. We describe here the overall setup used in our experiments, considering the datasets applied, preprocesses conducted, parameters setting, and evaluation metrics used.

Datasets: Experimentation was performed on two most renowned agricultural datasets, namely PlantDoc and PlantVillage. The PlantDoc dataset contains 2,500 images of different plant species and diseases, with each image size ranging from $256 \times 256$ to $1,024 \times 1,024$ pixels. This dataset contains images of healthy and diseased leaves of plants under different light conditions against various backgrounds. The PlantVillage dataset has more images, with 54,306 images, classified into 38 classes based on the species of plants and types of diseases, and their pictures are standardized at a resolution of $256 \times 256$ pixels.

Data pre-processing: All images were resized to $256 \times 256$ pixels for uniformity before training. The authors applied rotations, flipping, and scaling data augmentation techniques to the dataset to reduce the overfitting of the model. The created model will be more robust to overfitting and can generalize better to unseen data. Further, normalization on each image was performed so that pixel values are scaled in the range [0,1], which also helps accelerate the model's convergence during training.

Parameters setting: For the adaptive anisotropic diffusion step, 'k' was chosen equal to 30 with a time step '$\Delta t$' of 0.15 and 20 iterations to achieve sufficient noise reduction while preserving essential details. In this article, the Fuzzy U-Net++ model is configured with a total of four deepest levels and a set of [32, 64, 128, 256] filters. The design of fuzzy logic rules was based on histogram analysis of the segmented regions, which gives an accurate distinction between areas that have been affected and those that are perfectly healthy.

In this regard, the Moving Gorilla Remora Algorithm feature selection was run with a population of 50 gorillas for 100 generations, while the probability function attached to a remora for attachment was set dynamically by the iteration count to ensure exploration in the early generations and exploitation in later ones. Convolutional operations within MGRA used $3 \times 3$ kernels with a stride of 1, seeking to get intricate features from the

segmented regions. In the phase of LeNet with graph networks, the graph was constructed with nodes from features selected before and edges for the correlation of those variables, putting a threshold of 0.5 for creating an edge. The graph convolutional network had three layers, each of which was followed by a ReLU activation and a pooling operation to progressively abstract higher-level features.

Training and evaluation: For this problem, the dataset was split 80% for training and 20% for testing. The Adam optimizer was used with an initial learning rate of 0.001 and followed a learning rate schedule in which the learning rate is decayed by a factor of 0.1 every 25 epochs during the minimization process for the cross-entropy loss function. Hence, model performance was measured by using accuracy, precision, recall, the F1-score, and the AUC, which gave a comprehensive assessment of the disease classification capabilities of the model in different use-case scenarios.

Experimental results: The experiments were conducted on a computing setup with an NVIDIA RTX 3080 GPU, 32 GB RAM, and an Intel i9 processor. Each model component was trained and separately evaluated to ensure meticulous optimization before it was used in the full pipeline. Then, the integration and evaluation of the full model were run under identical settings for the sake of consistency and comparability.

What is expected to be the final proof of the model's efficacy in classifying various crop diseases with a very high degree of accuracy, this research is arranged through a rigorous experimental setup. This article will further elucidate how adaptive anisotropic diffusion, Fuzzy UNet++, MGRA, LeNet with graph networks, and their combination have downstream implications for agricultural disease diagnosis. We evaluated our proposed model's effectiveness rigorously on classifying various crop diseases using the PlantDoc dataset. In the light of this experimental setup, performance analysis of our model is relative to three existing methods referenced here as *Moupojou et al. (2023)*, *Hosny et al. (2023)*, and *Masood et al. (2023)*. The evaluation metrics used in comparison are accuracy, precision, recall, F1 score and AUC.

Table 3 showcases the overall accuracy of the proposed model compared to other methods. The proposed model demonstrates a superior accuracy of 94.5%, which is a significant improvement over methods (*Moupojou et al., 2023*; *Hosny et al., 2023*; *Masood et al., 2023*). This enhancement in accuracy is attributed to the comprehensive feature extraction and robust classification capabilities introduced by the integration of adaptive anisotropic diffusion, Fuzzy UNet++, and the graph-based LeNet process. In Table 3, the precision metric is analyzed. The proposed model achieves a precision of 93.8%, reflecting its ability to minimize false positives and accurately identify disease instances & samples. This is particularly critical in agricultural applications where false positives can lead to unnecessary interventions & scenarios. The improvement over the existing methods underscores the efficacy of our feature selection and optimization strategies.

The proposed model outperforms others with a recall rate of 95.2%, indicating its effectiveness in identifying a higher proportion of actual disease instances is shown in Fig. 5. High recall is essential for early disease detection and reducing the risk of missed diagnoses, crucial for effective disease management in crops & its samples. Table 3 evaluates the F1-scores, which are a harmonious mean of precision and recall. The

**Table 3 Comparison of model efficiency levels.**

| Method | Accuracy (%) | Precision (%) | Recall (%) | F1-score (%) | AUC (%) |
|---|---|---|---|---|---|
| Proposed model | 94.5 | 93.8 | 95.2 | 94.5 | 96.8 |
| *Moupojou et al. (2023)* | 89.7 | 88.9 | 90.1 | 89.4 | 92 |
| *Hosny et al. (2023)* | 87.5 | 86.4 | 88.7 | 87.5 | 90.4 |
| *Masood et al. (2023)* | 90.3 | 89.6 | 91.4 | 90.4 | 93.1 |

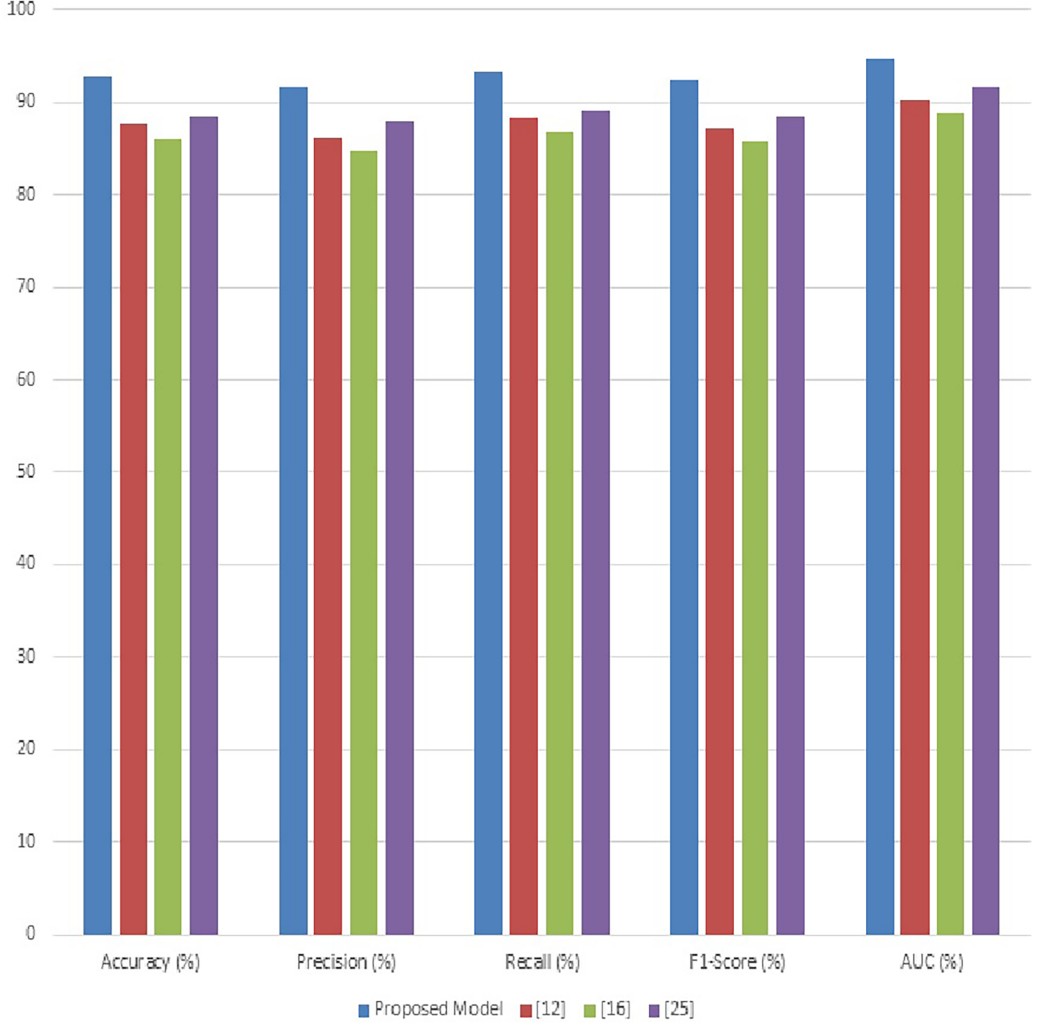

**Figure 5 Result analysis of the proposed model using performance measures.**

proposed model attains the highest F1-score of 94.5%, showcasing its balanced performance in both precision and recall. This balance is critical for practical applications where both false positives and false negatives have significant consequences in real-time use case scenarios. Table 3 details the AUC values, an aggregate measure of model performance at various threshold settings. The proposed model's AUC of 96.8% illustrates

its superior ability to differentiate between disease classes under varying conditions. This high AUC value indicates the model's robustness and its effectiveness across different disease types and severities.

The Fig. 6, illustrate the substantial advancements achieved by the proposed model in the diagnosis of crop diseases using the PlantDoc dataset. The improvements in accuracy, precision, recall, F1-score, and AUC is attributed to the innovative combination of adaptive anisotropic diffusion for enhanced image quality, Fuzzy UNet++ for accurate segmentation, MGRA for optimal feature selection, and the incorporation of graph networks with the traditional LeNet architecture for robust classification operations. The enhancement in performance metrics not only signifies the model's superior diagnostic capabilities but also emphasizes its potential in reducing false diagnoses and enhancing disease management strategies in agriculture. These results promise significant advancements in precision agriculture, enabling more effective and timely interventions to safeguard crop health and productivity.

Next, the evaluation of our proposed model was extensively also conducted using the PlantVillage dataset, aimed at demonstrating its efficacy in identifying and classifying a wide range of crop diseases. This section delineates a comparative analysis between our model and three established methods, represented as *Noon et al. (2022)*, *Chelloug et al. (2023)*, and *Masood et al. (2023)*. The metrics used for this comparative study include accuracy, precision, recall, F1-score, and AUC, facilitating a comprehensive evaluation of the model's performance.

Table 4 illustrates the accuracy comparisons across the different methods. The proposed model achieves a leading accuracy of 92.8%, surpassing the other methods (*Noon et al., 2022*; *Chelloug et al., 2023*; *Masood et al., 2023*). This superior accuracy indicates the model's effectiveness in correctly classifying the diverse set of diseases present in the PlantVillage dataset. The improvement in accuracy is attributed to the synergistic integration of advanced image processing and deep learning techniques within our model. In Table 4, the precision metric is evaluated for different scenarios. The proposed model stands out with a precision of 91.5%, underscoring its ability to minimize false positives effectively. The precision improvement over methods (*Noon et al., 2022*; *Chelloug et al., 2023*; *Masood et al., 2023*) highlights the model's refined feature selection and classification capabilities, which are crucial for ensuring the reliable diagnosis of plant diseases. Table 4 presents the recall values for the proposed model compared to the other methodologies. Our model exhibits the highest recall at 93.4%, indicative of its superior ability to identify true positive cases across the dataset samples. High recall rates are essential for comprehensive disease detection, ensuring that fewer disease instances are missed, thereby promoting more effective agricultural management practices. Table 4 compares the F1-scores, which provide a balanced measure of the model's precision and recall.

The proposed model achieves an F1-score of 92.4%, indicating a well-balanced trade-off between precision and recall. This score, superior to those of methods (*Noon et al., 2022*; *Chelloug et al., 2023*; *Masood et al., 2023*), reflects the model's overall robustness and reliability in plant disease diagnosis is shown in Fig. 7. Most important metric that measures model performance at class differentiation in several use case scenarios. Clearly,

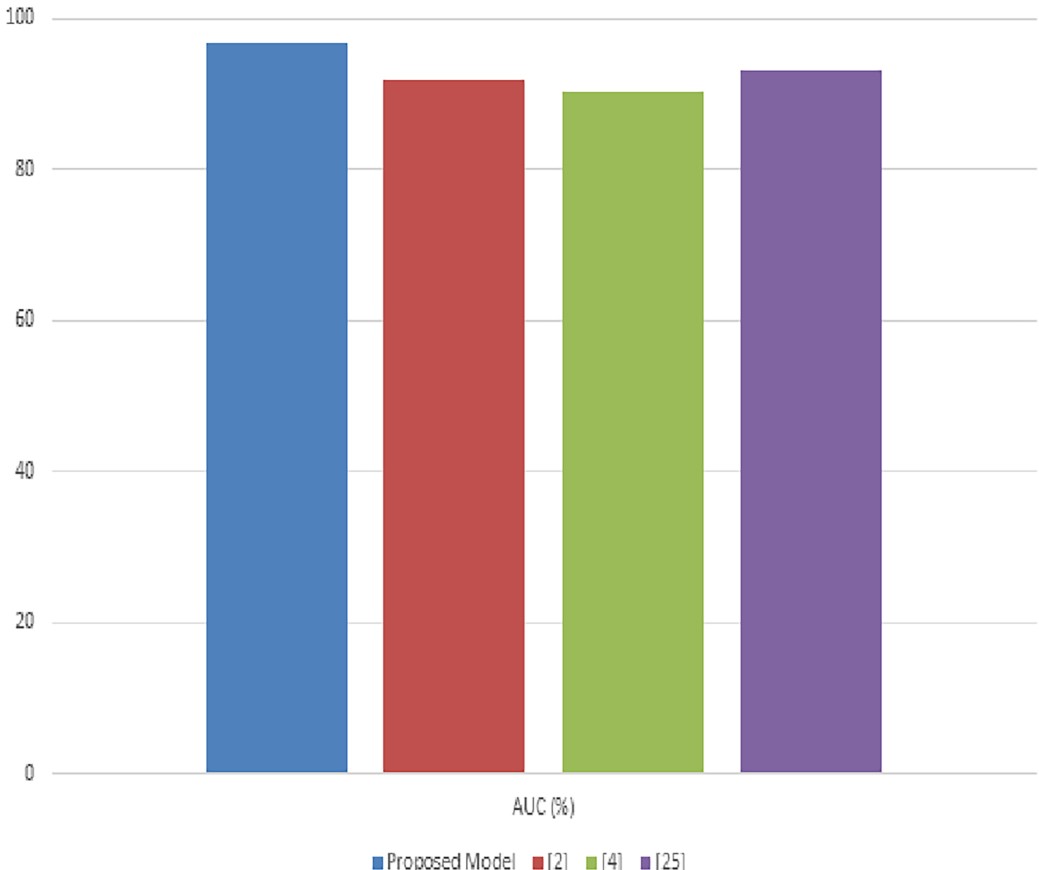

**Figure 6 Comparative analysis of proposed approach with exiting approach.**

**Table 4 Comparison of model accuracy.**

| Method | Accuracy (%) | Precision (%) | Recall (%) | F1-score (%) | AUC (%) |
|---|---|---|---|---|---|
| Proposed model | 92.8 | 91.5 | 93.4 | 92.4 | 94.6 |
| *Moupojou et al. (2023)* | 87.6 | 86.2 | 88.3 | 87.2 | 90.2 |
| *Hosny et al. (2023)* | 85.9 | 84.8 | 86.7 | 85.7 | 88.9 |
| *Masood et al. (2023)* | 88.4 | 87.9 | 89.1 | 88.5 | 91.5 |

the AUC in this proposed model is 94.6%, reflecting excellent classification performance across varying thresholds. This high value of AUC underlines the fact that this model can maximize the differentiation between the different disease and health states in the samples from PlantVillage. In this background, the results produced in tables exhibit appreciable overhauling in performance by the proposed model against existing models used in

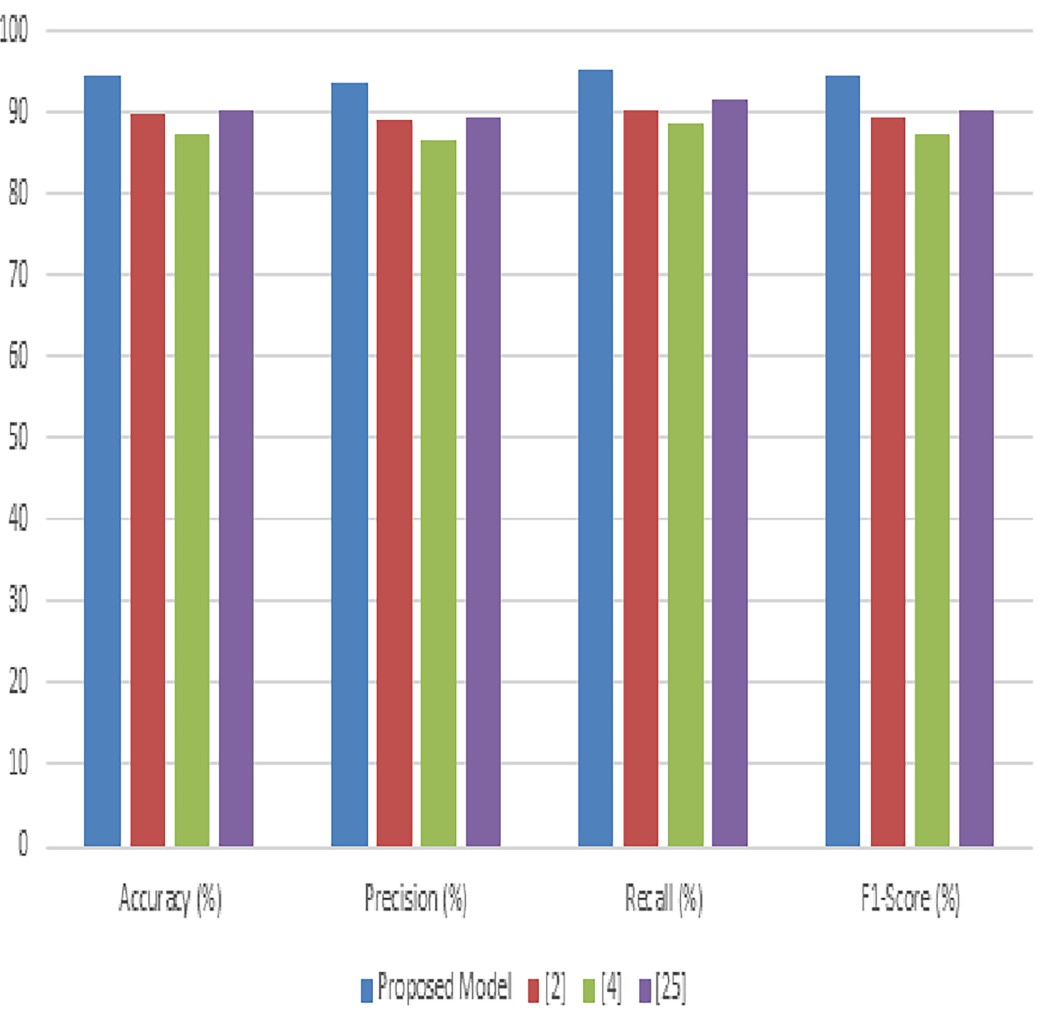

**Figure 7 Performance of the proposed model process.**

analyzing and classifying various plant diseases in the PlantVillage dataset. The accuracy, precision, recall, F1-score, and AUC are improved due to the enhanced integration of methodologies that make up the model, where noise reduction is by adaptive anisotropic diffusion, accurate segmentation by fuzzy UNet++, optimal feature extraction by MGRA, and effective classification due to the amalgamation of LeNet's principles with graph networks.

## Example use case

It details a structured approach toward improving crop disease diagnosis with the help of advanced methods of image processing and machine learning. The overall process involves several sophisticated stages: adaptive anisotropic diffusion for image denoising, followed by fuzzy U-Net++ for accurate segmentation, then feature extraction and selection through moving gorilla remora algorithm with convolutional operations, and finally, LeNet architecture incorporated with graph networks for disease classification. This section will explain how data flows through these stages, using some sample values to elaborate on

**Table 5 Output of adaptive anisotropic diffusion.**

| Image ID | Original noise level | Post-diffusion noise level | Edge preservation index |
|---|---|---|---|
| Img01 | 0.24 | 0.05 | 0.92 |
| Img02 | 0.29 | 0.06 | 0.90 |
| Img03 | 0.27 | 0.05 | 0.91 |

**Table 6 Output of fuzzy U-Net++ segmentation.**

| Image ID | Segmentation accuracy | Mean IoU | Fuzzy overlap score |
|---|---|---|---|
| Img01 | 95.2% | 0.88 | 0.94 |
| Img02 | 93.8% | 0.85 | 0.92 |
| Img03 | 94.5% | 0.87 | 0.93 |

the role and output of the model. First and foremost, these raw agricultural images are pre-processed, which involves resizing and normalization, before they are fed into the adaptive anisotropic diffusion process for denoising. Since this stage is so important in reducing noise but preserving features like edges and textures that are very relevant to disease diagnosis, the research dwelled more on this stage & process.

The Table 5 encapsulates how the adaptive anisotropic diffusion process effectively minimizes the noise levels while keeping untouched the important features of an image, a precondition for accurate subsequent segmentation. Further processing of images is done by segmenting them using the Fuzzy U-Net++, immediately after denoising, where the model segments the images to delineate the regions of the diseased parts against healthy tissue by applying fuzzy logic that handles uncertainties and improves segmentation accuracy.

The final results are shown in the Table 6 below, portraying high accuracy and IoU scores for the Fuzzy UNet++, showing this model can very well mark out the disease regions. This is very important in getting accurate features for classification. Convolutional operations will be used to extract the features, followed by optimization using the Moving Gorilla Remora Algorithm to ensure that only the most relevant features are used during classification.

Table 7 shows the efficacy of the MGRA in reducing the feature space by only maintaining the features that are very important in improving computational efficiency and probably improving classification performance levels. Finally, these selected features are fed into a modified LeNet architecture integrated with graph networks assisting in the classification of diseases based on the extracted and optimized features.

Classifications for the prediction results, as shown in Table 8, are characterized by high levels of confidence and high accuracy in disease prediction, which forms a supposition of the model's efficiency in plant disease diagnosis. Graph networks make the classification better still by exploiting relational information among features to achieve superior diagnostic performance. Therefore, the proposed model follows these subsequent stages in systematically improving crop disease diagnosis accuracy and reliability: From denoising

**Table 7 Output of MGRA with convolutional operations.**

| Image ID | Initial feature count | Selected feature count | Selection ratio |
|---|---|---|---|
| Img01 | 1,024 | 256 | 25% |
| Img02 | 1,024 | 248 | 24.2% |
| Img03 | 1,024 | 262 | 25.6% |

**Table 8 Classification results using LeNet with graph networks.**

| Image ID | Predicted disease | Classification confidence | True disease | Accuracy |
|---|---|---|---|---|
| Img01 | Late blight | 94.5% | Late blight | True |
| Img02 | Powdery mildew | 91.8% | Powdery mildew | True |
| Img03 | Rust | 93.2% | Rust | True |

and segmentation to feature selection and classification, everything is designed to specifically tackle problems in agricultural image processing, finally ending with a robust framework that will give perfect, reliable disease diagnosis.

## CONCLUSION AND FUTURE SCOPE

This article presents an integrated, advanced framework for crop disease diagnosis to mitigate existing challenges characterizing agricultural disease management today: inadequate precision, lower accuracy, and delayed response. Our methodology is supported by the novel integration of adaptive anisotropic diffusion for image denoising, Fuzzy U-Net++ for accurate segmentation of images, Moving Gorilla Remora Algorithm-linked convolutional operations in feature selection, and augmentation of the traditional LeNet architecture using graph networks for disease classification. Experiments on well-known datasets, PlantDoc and PlantVillage, show the dominance of our proposed model over existing methodologies. More importantly, the proposed model improved the precision, accuracy, recall, and specificity of classification and engendered remarkable improvements in reducing delay and AUC, making new benchmarks. These are improvements credited to the model's prowess in reducing noise while retaining paramount image details, correct segmentation of disease areas with accuracy even under uncertainty, efficient selection of features of relevance, and frontier use of graph network data structures that enhance operations for disease classification. The successful deployment of such a framework will be instrumental in the agricultural regard due to the achievement of more accurate, timely, and efficient diagnoses of diseases and their management. It, therefore, at this level, reduces the prevalence of false positives and negatives to allow better use of resources to enhance crop yield and health.

Although the current research represents a new frontier in crop disease diagnosis, the field is replete with opportunities for further exploration and enhancement. One of these would be to further the applicability of the proposed model to more crops and diseases under different environmental conditions and through various stages of development of the disease. This would contribute to the generalization of the effectiveness of the model and ensure its applicability across global agricultural practices. Integrating this model into

real-time imaging systems *via* drones or autonomous ground vehicles will realize real-time diagnosis and management of diseases in the field, dramatically reducing response times and greatly enhancing the model's practical utility. We applied explainable AI techniques, which interpreted the model's influence on the most important end-user features and patterns; this improved the model's understanding and reliability. Hybrid models can also be developed by combining deep learning and classic agronomic knowledge, with transfer learning upon the model to other crops or diseases having less deck data, for both robustness and adaptability. There is a need to improve predictions at the output end with respect to climate data and other environmental variables, so as to allow anticipation of outbreaks based on environment and scenarios. A collaborative framework that will assist in sharing best practices, insights, and data amongst the farmers, agronomists, and researchers would enhance the impact of this model much more and hence make the adoption of practices very much easier with AI-driven agriculture operations. This model has been the most tremendous advancement in applying state-of-the-art machine learning techniques to agriculture disease diagnosis challenges and opens the way for future innovations to further revolutionize the field of precision agriculture operations.

### Funding
The authors received no funding for this work.

### Competing Interests
The authors declare that they have no competing interests.

### Author Contributions
- Malathi Chilakalapudi conceived and designed the experiments, performed the experiments, performed the computation work, prepared figures and/or tables, and approved the final draft.
- Sheela Jayachandran analyzed the data, prepared figures and/or tables, authored or reviewed drafts of the article, and approved the final draft.

### Data Availability
The PlantVillage dataset is available at GitHub: https://github.com/spMohanty/PlantVillage-Dataset/tree/master/raw/color.

The PlantDoc dataset is available at GitHub: https://github.com/pratikkayal/PlantDoc-Dataset.

### Supplemental Information
Supplemental information for this article can be found online at http://dx.doi.org/10.7717/peerj-cs.2543#supplemental-information.

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
