# Peer review of "Iterative segmentation and classification for enhanced crop disease diagnosis using optimized hybrid U-Nets model"

_PeerJ Computer Science, doi:10.7717/peerj-cs.2543_

## Round 0.1 · original submission · Major Revisions

Dear authors,

Thank you for submitting your article. Feedback from the reviewers is now available. It is not recommended that your article be published in its current format. However, we strongly recommend that you address the issues raised by the reviewers, especially those related to readability, experimental design and validity, and resubmit your paper after the necessary changes and additions.

Best wishes,

·

Basic reporting

Manuscript ID Submission ID 102622V1
This paper is related to reviewing the manuscript titled " Iterative Segmentation and Classification for Enhanced Crop Disease Diagnosis Using Optimized Hybrid U-Net Model"

This study employs Adaptive Anisotropic Diffusion for image denoising, a Fuzzy UNet++ Model for enhanced image segmentation, and the Moving Gorilla Remora Algorithm (MGRA) combined with Convolutional Operations for optimal feature selection. Classification is inspired by the LeNet architecture, boosting disease identification accuracy.
Firstly, Although the proposed study is successful in terms of organization, presentation, content and results, major revision given in the following items need to be performed.

Experimental design

1) It would be better if the tables showing the comparison results of metrics such as AUC, recall, precision, accuracy were combined in a single table.
2) The mathematical model of Fuzzy UNet++ Model must be validated. Why is the recommended model a hybrid model? How is it strikingly different from Unet?
3) The leaf segmentation result can be shown more clearly. So, in this sense, segmentation of the diseased area in various regions on the leaf should be made. This can be done with a figure.
4) There may be some symbol errors in C(x,y,t) (Equation 4).
5) [2] or [12] reference given in Table 5? It should be checked.
6) In addition, the proposed model should be compared with more new CNNs or other deep learning methods.

Validity of the findings

.

Additional comments

My recommendation is major revision. I do not see any harm in publishing the manuscript once the above revisions are made.

Reviewer 2 ·

Basic reporting

-The technical contribution and experimental workload of this article deserve recognition. However, there are significant issues with the paper's writing. First and foremost, a majority of the citations are not formatted correctly, and both the introduction and methodology sections lack citations that explain the theoretical foundation and the reference backbone networks. The literature review's citations should not merely compile all existing methods into a table with the only 25 references in the entire paper. Additionally, there is a placeholder citation [] in line 214.
-The contribution and motivation of the article are relatively clear. However, the research background lacks sufficient literature support.

Experimental design

-The readability of the methodology section is poor. Appropriate subsections and other indicative markers should be established to enhance readability. For instance, separate the sections on Adaptive Anisotropic Diffusion, Fuzzy UNet++, and the Moving Gorilla Remora Algorithm.

Validity of the findings

-Figures 1 and 4 are compressed, resulting in poor visualization quality. Please ensure that their dimensions are not compressed.

-Figure 5 is overly simplistic; avoid using Excel for plotting as it lacks basic academic rigor.

-The benchmark experiments lack comparative visual result figures.

-The ablation study section lacks discussion and analysis of the results; the current manuscript only explains the meaning of each table.

·

Basic reporting

The paper demonstrates good quality of writing, with ample citations and a well-structured format.
However, since it introduces an Optimized Hybrid U-Net Model for image segmentation, the authors should specify the types of image backgrounds and content to which this method is applicable. Additionally, providing raw datasets with original images prior to segmentation would be beneficial for readers to better understand the performance of the proposed method across various scenarios.

Experimental design

From the presentation of formulas and code in the paper, it is clear that the methodology is described in detail with sufficient information, and the analysis is well-aligned with the characteristics of the research subject.
Nevertheless, in the "Review of Existing Methods" section, a more rigorous approach would involve categorizing these methods based on their contributions. The authors can first discuss the strengths and weaknesses of previous research, then explain the selection of baselines, methodological improvements, and experimental setups. For example, the motivation behind the design of image processing modules and parameter calculations in Figure 1 can be conveyed through such an organized approach.

Validity of the findings

The metrics used, including accuracy, precision, recall, F1-score, and the area under the curve (AUC), adequately reflect the model's performance.
Yet, readers and potential users of the proposed method might be more interested in understanding its applicability and practicality in different contexts, such as images with occlusions or shadows, and across various crop types. It is recommended that the authors provide more insightful comparisons and analyses in this area.

Additional comments

Clarity of charts could be improved.

---

## Round 0.2 · Minor Revisions

Dear authors,

Thank you for your paper. According to one reviewer, your paper still needs a minor revision and we encourage you to address the minor concerns and criticisms of Reviewer 1 and resubmit your article once you have updated it accordingly.

Best wishes,

·

Basic reporting

My recommendation is that the manuscript be accepted. Since the authors have completed most of the requested revisions, the following requested corrections have not been made.

1. "The performance results in Tables 3-9 should be combined into a single table"
2. "The leaf segmentation result can be shown more clearly. So, in this sense, segmentation of the diseased area in various regions on the leaf should be made. This can be done with a figure."

I see no problem in publishing the manuscript after these corrections are made.

Experimental design

My recommendation is that the manuscript be accepted. Since the authors have completed most of the requested revisions, the following requested corrections have not been made.

1. "The performance results in Tables 3-9 should be combined into a single table"
2. "The leaf segmentation result can be shown more clearly. So, in this sense, segmentation of the diseased area in various regions on the leaf should be made. This can be done with a figure."

I see no problem in publishing the manuscript after these corrections are made.

Validity of the findings

as above

---

## Round 0.3 · accepted · Accept

Dear Authors,

Thank you for the revised paper. I believe that the manuscript has been sufficiently improved and is now ready for publication.

Best wishes,

·

Basic reporting

The authors have made the requested corrections and taken the suggestions into consideration in the first round of revision . Therefore, I kindly request that this manuscript be accepted for publication in this journal.

Experimental design

Segmentation results are given .

Some results are combined in the Table 4

Validity of the findings

As above